



# Calibrating soybean parameters in JULES 5.0 from the US-Ne2/3 FLUXNET sites and the SoyFACE-O₃ experiment

Felix Leung[1,3], Karina Williams[2,7], Stephen Sitch[1], Amos P.K. Tai[3,6], Andy Wiltshire[2], Jemma Gornall[2], Elizabeth A. Ainsworth[4], Timothy Arkebauer[5], David Scoby [5]

[1] College of Life and Environmental Sciences, University of Exeter, Exeter, EX4 4RJ, UK.

[2] Met Office Hadley Centre, FitzRoy Road, Exeter, Devon, United Kingdom

[3] Earth System Science Programme, Faculty of Science, and Institute of Environment, Energy and Sustainability, The Chinese University of Hong Kong, Hong Kong

[4] USDA ARS, Global Change and Photosynthesis Research Unit, Urbana, Illinois, USA

[5] Department of Agronomy and Horticulture, University of Nebraska-Lincoln, Lincoln, Nebraska, USA

[6] State Key Laboratory of Agrobiotechnology, The Chinese University of Hong Kong, Hong Kong

[7] Global System Institute, University of Exeter, Laver Building, North Park Road, Exeter EX4 4QE, UK

*Correspondence to:* Felix Leung (felix.leung@cuhk.edu.hk)

**Abstract.** Tropospheric ozone ($O_3$) is the third most important anthropogenic greenhouse gas. $O_3$ is detrimental
to plant productivity, and it has a significant impact on crop yield. Currently, the Joint UK Land Environment Simulator (JULES) land surface model includes a representation of global crops (JULES-crop), but does not have crop-specific $O_3$ damage parameters, and applies default C3 grass $O_3$ parameters for soybean that underestimates $O_3$ damage. Physiological parameters for $O_3$ damage in soybean in JULES-crop were calibrated against leaf gas-exchange measurements from the Soybean Free-Air-Concentration-Enrichment (SoyFACE) with $O_3$ experiment
in Illinois, USA. Other plant parameters were calibrated using an extensive array of soybean observations such as crop height, leaf carbon, etc. and meteorological data from FLUXNET sites near Mead, Nebraska, USA. The yield, aboveground carbon and leaf area index (LAI) of soybean from the SoyFACE experiment were used to evaluate the newly calibrated parameters. The result shows good performance for yield, with the modelled yield being within the spread of the SoyFACE observations. Although JULES-crop is able to reproduce observed LAI
seasonality, its magnitude is underestimated. The newly calibrated version of JULES will be applied regionally and globally in future JULES simulations. This study helps to build a state-of-the-art impact assessment model and contribute to a more complete understanding of the impacts of climate change on food production.



## 1 Introduction


Surface ozone ($O_3$) pollution is one of the major threats to global food security due to the detrimental effects of ozone exposure on crops (Ainsworth et al., 2012; Avnery et al., 2011b; Leung et al., 2020; Long et al., 2005; Tai et al., 2014; Tai and Val Martin, 2017). In the United States alone, crop loss due to tropospheric $O_3$ costs more than \$5 billion USD annually (Ainsworth et al., 2012; Avnery et al., 2011a; Van Dingenen et al., 2009).


Soybean is one of the main staple crops for human consumption; it also serves as an important source of animal feed. It is a cheap source of proteins and therefore soybean products are consumed around the world. The impact of $O_3$ on soybean physiology and growth has been studied extensively (Ainsworth et al., 2012; Betzelberger et al., 2012; Dermody et al., 2008; Morgan et al., 2003). Crop yield losses to tropospheric $O_3$ have been quantified using

model projection and experiments. The National Crop Loss Assessment Network and European Open Top Chamber programs have established the air quality guideline, which derived dose-response relationships from comparable experimental data. These campaigns provided critical information such as the $O_3$ response relationship and estimated yield loss due to $O_3$ damage that enabled regional projections of $O_3$ effects on crop yields (Fuhrer, 2009). However, open top chambers modify plant response to $O_3$ due to the 'chamber effects' which create

microclimates (Elagöz and Manning, 2005) and environmental differences between the chamber and open air micrometeorology in which yield loss is underestimated (Van Dingenen et al., 2009). Recently the introduction of Free-Air-Concentration-Enrichment (FACE) technology avoids the artefacts from enclosed chambers, and $O_3$ fumigation was adapted to FACE facilities (Morgan et al., 2004). The application of FACE experiment on crops took place in China (Zhu et al., 2011) and USA, including experiments with soybean at the

SoyFACE experiment in Champaign, Illinois (Morgan et al., 2004; Betzelberger et al. 2010; 2012).

Crops are a significant component of the land surface; e.g., croplands and pasturelands represent 12% and 26% of the global terrestrial land, respectively (Van den Hoof et al., 2011). Moreover, the phenology of crops is very different from that of natural vegetation, and is characterized by high growth, turnover rate, and strong

seasonality. It is thus necessary to include a crop-specific parameterization scheme to improve simulations of land surface fluxes and regional climate in agroecosystems (Van den Hoof et al., 2011). The Joint UK Land Environment Simulator with crops (JULES-crop) is a crop parameterisation (Osborne et al., 2015) within the land surface model, JULES (Best et al., 2011; Clark et al., 2011). Global simulations have been performed with JULES-crop for rice, wheat, maize and soybean (Osborne et al., 2015). These four crop types contribute more



than 70% of human calorie intake (Ray et al., 2013). JULES-crop includes routines representing growth, development and harvesting of crops driven by the overlying meteorological inputs. In JULES-crop, four new prognostic variables have been added: crop development index (DVI), root carbon (Croot), harvest carbon (Charv) and reserve carbon (Cresv). DVI controls the duration of the crop growing season in four distinct stages: sowing, emergence, flowering, and maturity, and it determines when changes in carbon partitioning occur

(Osborne and Hooker, 2011). Croot, Charv, and Cresv are the carbon pools for roots, harvested organs (e.g. grains of cereal, fruits, and root) and stem reserves, respectively. Carbon pools for stem and leaves are determined from the existing prognostic variables, LAI (Leaf area index) and canopy height. In Osborne et al. (2015), global runs of maize, wheat, soybean and rice were carried out using JULES-crop. Site runs were performed at four FLUXNET sites with soybean-maize rotation: Bondville (US-Bo1), Fermi (US-IB1) and Mead (US-Ne2 and

US-Ne3). Simulated yield was compared against country and global FAO crop yields.  Osborne et al. (2015) used generic representations for each of the crops in their global study. For the plant parameters that are needed outside the crop model such as leaf nitrogen and leaf respiration parameters, these are set to those of the C3 or C4 grass functional types. Osborne et al. (2015) suggested that these parameters could be tuned to be more crop specific to improve fit to observations. These JULES parameters have been calibrated against observations for

maize, using data from the Mead FLUXNET sites in Nebraska (Williams et al., 2017). However, to date, these parameters have not been calibrated to soybean data.

There are many crop models developed by institutions / organisations around the world. Most are designed for application to an individual field up to the regional scale and do not include $O_3$ impacts on vegetation.

(Appendix Table A1) compares a selection of land surface models which include crop tiles and have the functions to model climate impact on crop productivity. JULES-crop is of particular interest because it is a development of the global land surface component JULES of the Met Office numerical weather prediction and climate models, and contains a detailed representation of plant physiological processes at sub-diurnal timescales, including consideration of $O_3$ effects on natural vegetation, thus making it suitable for this study.

JULES-Crop has been accepted into the JULES trunk with the intention to be coupled with the Hadley Centre Global Environment Model (HadGEM) in the near future. HadGEM is recognised as one of the best performing climate models with smaller errors than typical climate models (Gleckler et al., 2008; Knutti et al., 2013).






The calibration of O₃ damage on soybean would allow land surface and crop models to more realistically and reliably simulate present-day and future O₃ damage, and subsequently to quantify its economic impacts. The objective of this study is to calibrate soybean representation for JULES-crop, with a particular focus on the response of soybean to O₃ exposure.


This paper is organised as follows: Section 2 describes the model set-up and observations used for the JULES calibration. Section 3 compares the results from the calibrated JULES runs against independent observations. Section 4 assesses the suitability of the model for modelling soybean under O₃ damage and discusses ways of future model improvement.


## 2 Methods

A flowchart demonstrating the calibration and evaluation procedure is given in Figure 1. We first tuned the JULES-crop soybean parameterisation at the US-Ne2 and US-Ne3 Mead sites, where ten years of soybean

physiological and meteorological observations were available, at ambient ozone (Figure 1, steps 1-5).

Secondly, to calibrate the JULES ozone damage parameters (Figure 1, step 6) we made the assumption that there is a negligible damage to crop yield at ambient background levels of O₃ at both the SoyFACE and Mead sites. This is consistent with Mills et al. (2007), who reviewed over 700 published papers and conference proceedings

and found that O₃ level of AOT40 over 3 months of 5 ppm-h reduced soybean yield by less than 5%. Then we calibrated specifically the soybean O₃ response using leaf gas exchange measurements from soybean grown under elevated O₃ concentrations at SoyFACE.

Finally, we applied JULES-crop newly calibrated for soybean and its O₃ sensitivity at the leaf-level and evaluated

model performance against observed yield and leaf area index from SoyFACE, taken for the full range of rings and cultivars (Figure 1, step 7).

### 2.1 Calibration of soybean in the absence of ozone damage, using observations from Mead

We followed the standard tuning procedure performed on maize by Williams et al., (2017) but applied to soybean (Figure 1, steps1-5). This method is described in detail in the Supplementary Material, and the resulting



parameters are given in Table 1-3. These are compared to the parameters used in Osborne et al. (2015), which we refer to as the "Osborne 2015 tuning". Note that the parameters in Table 3 in the Osborne 2015 tuning are typical defaults for C3 grass, rather than soybean-specific.


## 2.2 Calibration of JULES ozone damage parameters

### 2.2.1 Ozone effects on vegetation (exposure-response)

Many studies have shown that the impacts of $O_3$ are closely related to accumulated exposure above a threshold concentration rather than the mean growing season concentration (Forestry Commission, 2016; Gerosa et al. 2012; Mills et al. 2007). An index of accumulated exposure above a threshold concentration of x ppb (AOTx) has thus been developed as a measure of assessing $O_3$ pollution effects on vegetation. AOTx is calculated as the summed product of the concentration above the threshold concentration and time (T), with values expressed in ppb h or ppm h. (Mills et al. 2007; Forestry Commission, 2016).

The $O_3$ exposure index AOT40: Accumulated $O_3$ exposure over a threshold of 40 parts per billion (Equation 1) has been widely used by crop impact models in the forestry and agriculture industry and was used at SoyFACE.

$$AOT40 = \int \max(O_3 - 40ppb, 0.0)\, dt \qquad (1)$$

The metric ensures only $O_3$ concentrations above 40 ppb are included. The integral is taken over daytime hours. AOT40 does not account for the actual uptake of $O_3$ by plants and how this varies with ontogenetic (life span of the plant) and climatic factors such as temperature, irradiance, vapour pressure deficit, and/or soil moisture (Ashmore, 2005; Fuhrer et al. 1997).

There is a drawback of the cumulative $O_3$ exposure indices (Pleijel et al. 2000), which assume an instantaneously fixed threshold flux below which there is no effect of $O_3$, which may not be realistic. Also in nature, the threshold value is unlikely to be constant (Ashmore, 2005) since the capacity of detoxification of $O_3$ varies with climate and plant species. To improve these indices, the Stockholm Environment Institute developed the Deposition of Ozone for Stomatal Exchange model (DO3SE) (Emberson et al., 2007). DO3SE was developed to estimate the risk of $O_3$ damage to European vegetation and is capable of providing $O_3$ flux estimation by evaluating the soil water deficits and their influence on stomatal conductance which affect plant $O_3$ uptake. Phyto-toxic $O_3$ dose (POD) above a stomatal threshold over a growing season (the accumulated stomatal flux above threshold Y) PODy can differentiate species sensitivity to rising background concentration, while AOT40 can only incorporate the effect



of rising global background $O_3$ above the threshold 40ppb. This difference means AOT40 metric is less sensitive to $O_3$ peaks, and stomatal flux based metric (e.g. PODy and DO3SE) perform better on $O_3$ damage estimation in general (Büker et al., 2012; Dentener, F., Keating, T., and Akimoto, 2010; Pleijel et al., 2007).


### 2.2.2 Description of ozone response scheme in JULES

The current $O_3$ scheme in JULES uses a dose-response approach to model $O_3$ damage (Sitch et al., 2007; Clark et al., 2011). It uses the $O_3$ concentration in the atmosphere to modify net photosynthesis $A_p$ by an $O_3$ uptake factor $f$:

$$A = A_p f \quad (2)$$

where $f$ represents the fractional reduction of plant production:

$$\mathbf{f} = 1 - aUO_{>FO3crit} \quad (3)$$

It assumes that $O_3$ suppresses the potential net leaf photosynthesis in proportion to the $O_3$ flux through stomata above a specified critical threshold (Clark et al., 2011).

$UO_{>F\ O3crit}$ is the instantaneous leaf uptake of $O_3$ over a plant functional type specific threshold (FO3crit) (nmol m$^{-2}$s$^{-1}$) and the plant type specific parameter $a$ is the fractional reduction of photosynthesis with $O_3$ uptake by leaves (Clark et al., 2011; Sitch et al., 2007).

$$UO_{>FO3crit} = \max[(\,F_{O3} - F_{O3crit}), 0.0] \quad (4)$$

From equations 3 & 4, $F$ depends on the $O_3$ uptake rate by stomata ($F_{O3}$) over a critical (plant functional type specific) threshold for damage. It uses an analogy of Ohm's law, the $O_3$ flux through stomata, $F_{O3}$ (nmol $O_3$ m-2 s-

1), is given by,

$$F_{O3} = \frac{[O_3]}{R_a + \left[\frac{\kappa_{O3}}{gl}\right]} \quad (5)$$





where [$O_3$] is the molar concentration of $O_3$ at reference level (nmol m-3), $R_a$ is the combined aerodynamic and

boundary layer resistance between leaf surface and reference level (s m-1). $g_l$ is the leaf conductance for $H_2O$ (m

s-1), and $\kappa_{O3}$ = 1.67 is the ratio of leaf resistance for $O_3$ to leaf resistance for water vapour [Sitch et al., 2007]. The

uptake flux is dependent on the stomatal conductance, which is reliant on the photosynthetic rate in JULES. Given

that $g_l$ and photosynthetic rate $A$ are linear related [Cox et al., 1999], $g_l$ is given by,

$$g_l = g_p f \qquad (6)$$


Where $g_p$ is the leaf conductance in the absence of $O_3$ effects. The set of equations (3,5,6) produces a quadratic

relationship as a function of $f$, that can be solved analytically (Sitch et al., 2007).

Fractional reduction of photosynthesis with the instanteneous uptake of $O_3$ by leaves (mmol m-2) (dfp_dcuo_io)

determines the sensitivity of soybean to $O_3$ and the PFT-specific $O_3$ critical level (FO3 crit) determines the

threshold $O_3$ flux which would cause damage to photosynthesis (Oliver et al., 2018; Sitch et al., 2007). The higher

the sensitivity of plants to $O_3$ the lower photosynthesis the plant has at a given constant critical threshold. Sitch et

al. (2007) configured plant functional types with two different $O_3$ sensitivities (fractional reduction of

photosynthesis by $O_3$, $F$, equation 1, 2), where $F$ = 1.40 is high sensitivity, and $F$ = 0.25 is lower sensitivity for

C3 grass (Sitch, 2007), using monthly average $O_3$ data and calibration to yield observations.

### 2.2.4 Calibrating the ozone effects on crop leaf photosynthesis in JULES using SoyFACE

The SoyFACE experiment in Illinois allows controlled $CO_2$ or $O_3$ enrichment across large plots within a soybean

field without an enclosure. SoyFACE $O_3$ fumigation typically began after the emergence of soybean, and the plots

were fumigated with $O_3$ for 8–9 hours daily except when leaves were wet. In 2009 and 2010, soybeans were

exposed to nine different concentrations of $O_3$ ranging from the ambient level to a target level of 200 ppb (Figure

A2). The fumigation ended when soybean was mature.

Plant damage from $O_3$ is cumulative and the target concentration for the experiment was not always met (e.g.,

when wind speeds are low, during rain or when $O_3$ generators or analyzers are down). Therefore, the 8-hour mean

and the AOT40 index (Accumulated Ozone exposure above the Threshold of 40 ppb) were used for the analysis

in SoyFACE instead of using the target $O_3$ concentration. The planting dates were June 6, 2009 (Day 159) and

May 27, 2010 (Day 157). Fumigation began on June 29, 2009 (Day 179 - 260) and June 6, 2010 (Day 167 -271)

and harvest occurred on October 20, 2009 (Day 293) and September 20, 2010 (Day 273). $O_3$ concentrations



measured at SoyFACE fluctuated greatly, as they were strongly influenced by weather conditions, especially by wind speed. The magnitude of $O_3$ concentration fluctuations in the high targeted concentration was greater than the low concentration (Figure A2). On some days of the year when the fumigation was off, very low $O_3$ concentrations were recorded for all target rings.

To calibrate the $O_3$ parameters for soybean in JULES-crop, we used midday photosynthetic gas-exchange measurements from Betzelberger et al. (2012). These were taken at four stages during the growing season, from seven soybean cultivars growing at 9 different $O_3$ concentrations, using open gas exchange systems (LI-6400 and LI-6400-40). These observations were used in conjunction with the daytime 8-hour mean $O_3$ concentration measurements and the parameters calibrated at the Mead site to drive the Leaf Simulator computer package, which

reproduces the calculation of leaf photosynthesis within JULES. We then tuned the $O_3$ parameterisation of Fractional reduction of photosynthesis by $O_3$ (sensitivity) and Threshold of $O_3$ flux (mmol m-2 s-1) to match the modelled leaf photosynthesis rate to the observed rate (Figure 2). The tuned parameters are showed in Table 4.


### 2.3 Model configuration for the JULES-crop SoyFACE runs

The meteorological forcing data measured at Champaign, Illinois in 2009 (Ainsworth et al., 2010) were used to drive the JULES-crop model. The downward longwave radiation and diffuse radiation data from NOAA at

Bondville site (SURFRAD) were used as SoyFACE does not have these variables available. The driving data were repeatedly applied (recycled 25 times) to spin up the model from an arbitrary starting point with soil temperature initially set to 278 K and soil moisture to 75% of saturation. A single crop type was modelled – soybean – using a single plant tile. Observed $CO_2$ (NOAA) and 8-hour mean observed $O_3$ concentrations from the SoyFACE rings (averaged over a month) were used as the driving data of the model since natural $O_3$ is produced around 8 hours

in daytime and it is a typical temporal resolution for $O_3$ fumigation. The soil ancillary parameters used in SoyFACE were extracted from the global dataset of soil ancillary from the HadGEM2-ES model (a coupled Earth System Model that was used by the Met Office Hadley Centre for the CMIP5). Observed ambient $O_3$ were used as the control. The new parameters for soybean were used, which we calibrated to observations from the Mead FLUXNET sites as described in the supplementary material. The exception is the initial carbon: since the row

spacing at the SoyFACE experiment is half that used at the Mead sites, we doubled the initial carbon for SoyFACE



compared to Mead. The resulting model yield, above ground carbon and LAI was compared to the SoyFACE observations.

## 3 Results and Discussion

### 3.1 Results from JULES runs with crop model and ozone damage turned on

Figure 3 shows the evaluation of the soybean aboveground biomass carbon for different $O_3$ exposure levels (AOT40) using the $O_3$ damage parameters in Table 4. The model aboveground carbon (solid lines) are compared to the line fitted in Betzelberger et al 2012 to their aboveground carbon observations. The run with the newly-calibrated parameters overestimated the carbon at ambient ozone levels. One contributing factor could be that water stress is underestimated in the new configuration, since it was not possible to evaluate the response to soil water availability using the Mead site data, so we instead derived a value for p0 (parameterise in the calculation of the threshold for water stress, see Table 3) from literature. We tested the sensitivity to this choice by re-running this configuration with p0=0, and this caused a 12% reduction in aboveground carbon (plots not shown). In addition, the representation of the soil properties in the JULES SoyFACE run could be improved by calibration to site measurements. In contrast, the "Osborne 2015 tuning" intersects the line fitted to observed aboveground carbon at zero ozone concentration (partially because of higher water stress), but then shows a sharp decrease from zero to ambient levels, which is not realistic. Note that no observations were taken for below-ambient ozone concentrations at SoyFACE, so this section of the fitted line is an extrapolation. The slope of the aboveground carbon response to increasing ozone concentrations is similar for all three runs, and compares very well to the Betzelberger et al 2012 fitted line.

The yield-$O_3$ response curve in Figure 4 show that new parametrisation slightly overestimates yield in the ambient SoyFACE ring, compared to the spread of SoyFACE yield observations from Betzelberger et al 2012. The 'Osborne 2015 tuning' with high ozone sensitivity is within the spread of measured yield in ambient conditions, but note that the modelled yield has decreased sharply from zero ozone concentration to ambient levels, which is undesirable. The magnitude of the gradient of yield against AOT40 for all three model configurations is within the spread of the observations. However, the slope is underestimated for the new, calibrated run and overestimated for the 'Osborne 2015 tuning', especially for the range from ambient to 40 ppm h. Recall that ozone concentration modifies net leaf $CO_2$ assimilation rate in JULES, and that the model parameters governing this process (*Fo3crit,*



*a*) are calibrated directly to net leaf $CO_2$ assimilation rate observations from SoyFACE in our new configuration (Section 2). Reductions in the modelled net leaf $CO_2$ assimilation rate lead to the reductions in model aboveground biomass, yield and LAI which we show in this section. However, Betzelberger et al 2012 also reported additional

impacts of ozone damage, such as changes in leaf absorptance and specific leaf mass, that are not represented in JULES, and therefore our tuning does not account for them. In contrast, the values of *Fo3crit* and *a* in the high and low sensitivity versions of the 'Osborne 2015 tuning' simulations (table 4) were calibrated in Sitch et al 2007 to yield observations. Therefore, they can be seen as 'effective' parameters in these configurations, since they incorporate the effect of the ozone damage processes that are not explicitly represented in JULES.


.

Note that we plot AOT40 on the x-axis for illustrative purposes only, to be comparable with results presented in Betzelberger et al 2012 - AOT40 was not used in the JULES run. An alternative would be to plot ring number or ring target concentration. Ideally, we would plot the x-axis with the metric Phytotoxic Ozone Dose (POD) for

JULES and observed data, which account the dosage of $O_3$ that get into the stomata of soybean, but is beyond the scope of the present study.

Figure 5 compares the model and observed LAI at SoyFACE for different $O_3$ concentrations. JULES was able to reproduce LAI seasonality; however, it underestimated the amplitude. The maximum LAI for calibrated JULES

peaked around day 240 in September and observations peaked at DOY 220~230. The peak LAI in the model runs was less than half the observed LAI in all cases. While the Mead model runs also showed a slight underestimation of peak LAI compared to observation (Supplementary Materials), the majority of the underestimation of the modelled SoyFACE LAI is due to a difference between the observed relationships between peak LAI and yield at the Mead and SoyFACE sites. At both sites, observed maximum yield increases with observed peak LAI.

However, for similar observed yields, the observed SoyFACE yield tends to be higher than the observed Mead LAI. Given that our calibration is based on Mead observations, it is therefore not surprising that our model runs at SoyFACE underestimate peak LAI compared to the SoyFACE observations.

A contributing factor to the different relationship between observed peak LAI and observed yield at SoyFACE

compared to Mead could be the different methods used to measure LAI at the Mead sites (which this parameter set was tuned against) and at SoyFACE. At Mead, destructive measurements were taken, whereas at SoyFACE, LAI was measured indirectly, using radiation attenuation through the canopy.





Another plausible contributing factor for the different relationship between observed peak LAI and observed yield
at SoyFACE compared to Mead is the row density of the soybean. The SoyFACE row spacing was half that of
Mead so, as described above, we set the initial carbon to twice that observed at Mead. The denser planting allowed
soybean at SoyFACE to reach higher LAI earlier in the growing season. If this also resulted in thinner leaves at
the beginning of the season than with the Mead row spacing, then this could explain the difference in the peak
LAI to yield relationship between the two sites. JULES also does not account for leaf age on leaf assimilation rate
- in reality a lower leaf assimilation is observed in the late season associated with leaf aging, and it is plausible
that this could also be affected by row spacing.

Figure 5 also demonstrates that model LAI responds more to ozone concentrations than the observed LAI. One
contributing factor is the observed decrease in specific leaf area at SoyFACE in increased ozone (Betzelberger et
al 2012). As mentioned above, this process is not captured by JULES. This issue is particularly pronounced in the
Osborne 2015 tuning runs, where the modelled LAI in the ring with target 200ppb is roughly a third of the peak
LAI in the ambient ring.

## 5 Conclusions

Climate change and air pollution are a great threat to food production. JULES-crop has been developed to
represent crops in the land surface model and allow us to estimate the future climate and air pollution impact to
crops. The $O_3$ impact on crops could be quantified with an improved parameterization to the existing $O_3$ damage
scheme for C3 plants. The default soybean biochemical and respiratory parameters in JULES were based on C3
grass parameters. Characteristics of soybean are more similar to a shrub than grass, therefore parameter calibration
is needed to improve the performance of soybean in JULES-crop.

In this paper, the parameters needed to describe soybean in JULES-crop were first revised against observations
from the Mead FLUXNET sites to ensure that the crop, biochemical and respiratory parameters explicitly
represented soybean. Compared with observations from these sites showed that GPP and LAI were well
represented for irrigated soybean at Mead. The $O_3$ damage parameterisation was subsequently calibrated against
leaf gas exchange observations from the Soybean Free-Air-Concentration-Enrichment (FACE) experiment for the
$O_3$ damage, by tuning the sensitivity and critical threshold of $O_3$ damage. On the whole, JULES-Crop reproduces



the observed negative correlation between yield and O₃ exposure. It also reproduced the negative impacts of ozone

on LAI, and the seasonality of phenology, although the simulated LAI was underestimated at SoyFACE. This
method of calibrating soybean could be replicated for other crops once data become available and would contribute
to more accurate parameters for crop models. The calibration will be applied to a regional and transient run and
eventually the newly calibrated JULES-crop for soybean and its sensitivity to O₃ damage, coupled within an Earth
System Model.


*Code availability*. This study uses JULES version 5.0 releases. The code and configuration for the SoyFACE
runs can be downloaded via the  Met Office Science Repository Service (MOSRS) at
https://code.metoffice.gov.uk/trac/roses-u/browser/a/r/8/6/6/trunk (JULES Collaboration, 2018)(registration

required) and are freely available subject to completion of a software licence. The Leaf Simulator can be
downloaded from https://code.metoffice.gov.uk/trac/utils (Williams et al., 2018) (login required).

*Data availability*. Unless otherwise noted, all site observations discussed in this paper were obtained from the
Site Information pages of the AmeriFlux website hosted by Oak Ridge National Laboratory
(http://fluxnet.fluxdata.org/,(AmeriFlux collaboration, 2018)) or by personal communication with the Mead sites
Research Technologist. The longwave radiation, diffuse radiation and air pressure from Bondville, Illinois site is
obtained by the SURFRAD (Surface radiation) network from

ftp://aftp.cmdl.noaa.gov/data/radiation/surfrad/Bondville_IL/. The SoyFACE data used for the run are available
on MOSRS at:
https://code.metoffice.gov.uk/trac/roses-u/browser/a/r/8/6/6/trunk/driving_data
https://code.metoffice.gov.uk/trac/roses-u/browser/a/r/8/6/6/trunk/bin/SoyFACE_gas_exchange_data_2009.csv
https://code.metoffice.gov.uk/trac/roses-u/browser/a/r/8/6/6/trunk/ancil_data

Accessing the MOSRS requires registration, but once you access into the system, there's no information about
who is downloading or viewing which pages.

*Acknowledgements*. Felix Leung gratefully acknowledges financial support from the NERC CASE Studentship
with Met Office (NE/J017337/1), "Impact of tropospheric O₃ on crop production under future climate and



atmospheric CO2 concentrations, and their interactions within the Earth System". Karina Williams gratefully acknowledges financial support from the European Commission under grant agreements 308291 (EUPORIAS), 603864 (HELIX). We acknowledge the following AmeriFlux sites for their data records: US-Ne1, US-Ne1, US-Ne3. In addition, funding for AmeriFlux data resources and core site data was provided by the U.S. Department of Energy's Office of Science.



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





**Step 3: Tune parameters needed by JULES-crop only:**

- Crop height vs stem biomass
- Sowing, emergence, flowering, harvest dates
- Air temperature time series
- Leaf carbon to leaf biomass ratio
- Seed fraction of harvest pool
- Time series of green leaf, yellow leaf, stem, harvest pod biomass
- Parameters in effective temperature
- Carbon to biomass ratio for stem, roots, harvest pods
- Root carbon compared to total plant carbon as a function of DVI

**Step 1: Tune parameters needed by all PFTs in JULES**

- Hourly GPP against APAR for LAI 3.5-4.5 (with dependence on soil moisture, VPD, diffuse radiation fraction and air temperature)
- Hourly FAPAR against LAI for diffuse fractions
- Ratio of leaf nitrogen to leaf carbon
- Respiration parameters
- Root, stem nitrogen to carbon ratios

Literature

Mead observation

SoyFACE observation

**Step 4: Demonstrate with Mead runs forced with a derived NPP** (calculated from change in carbon pods over time)

Compare model to observation:

- Leaf, stem, harvest pool biomass against DVI

**Step 2: Demonstrate with Mead runs forced with meteorology data, LAI, height** (compare model to observation)

- GPP against time

**Step 5: Demonstrate with full JULES-crop runs at Mead,** forced by Mead meteorology data

Compare model to observation:

- GPP, LAI, height, aboveground carbon, carbon in harvest pod against time

**SoyFACE site-specific data:**

- Met forcing (SURFRAD data for diffuse radiation fraction)
- Planting, emergence, flowering, harvest dates
- Latitude, longitude
- Soil properties
- Global $CO_2$ concentration for 2009 (NOAA)
- Ozone levels in each ring (monthly time series)

**Step 7: Run JULES-crop at SoyFACE**

Evaluate against yield, aboveground carbon and LAI for each ring

**Step 6: Tune JULES ozone damage parameters using:**

LiCOR measurements for each ring and cultivar


Figure 1. Flowchart of tuning the parameters and calibrating the model





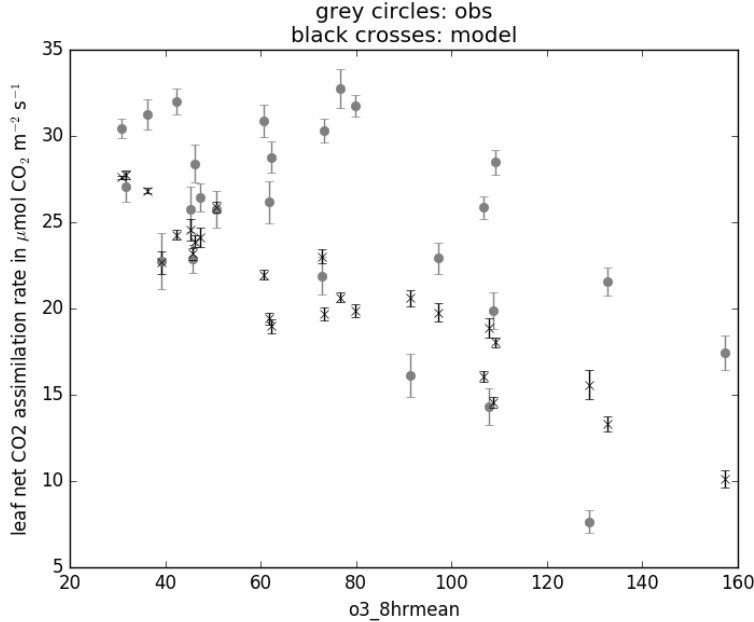


Figure 2. Net leaf $CO_2$ assimilation rate for calibrated JULES, simulated using the Leaf Simulator (black crosses) and observations from Betzelberger et al., (2012) (grey circles). X-axis is the daytime 8-hour mean $O_3$ concentration (ppb) .




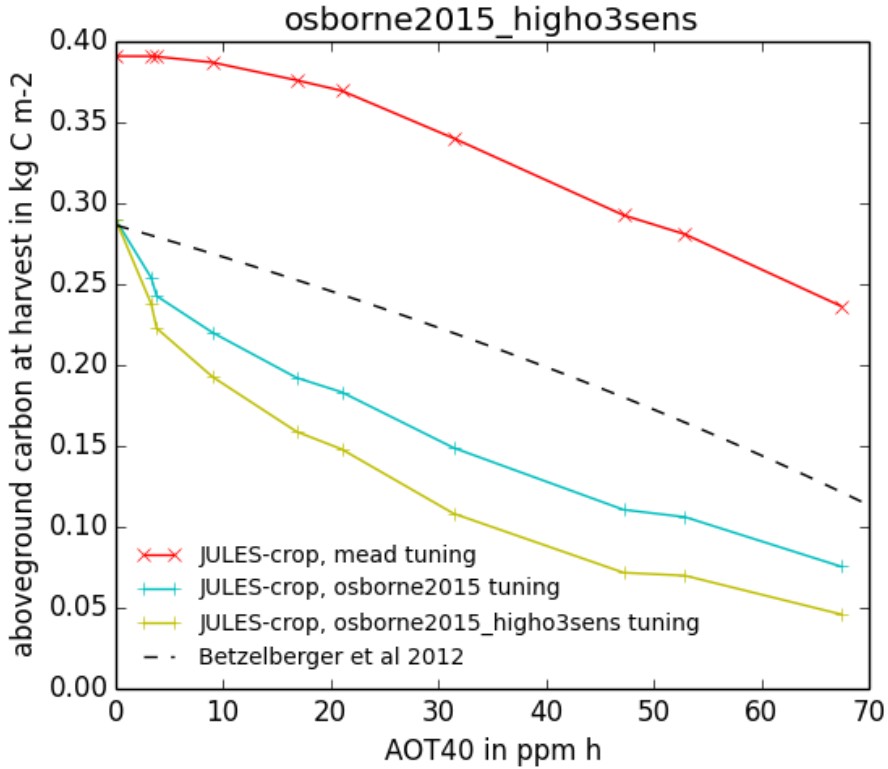

**Figure 3**. Aboveground carbon biomass of soybean at harvest stage for calibrated Joint UK Land Environment Simulator with Crop module turned on (JULES-crop) using the Mead soybean tuning (red), Osborne et al. (2015) standard parameters with Sitch et al. (2007) low ozone sensitivity (blue), high ozone sensitivity (green) and observation from SoyFACE from Betzelberger et al. 2012.




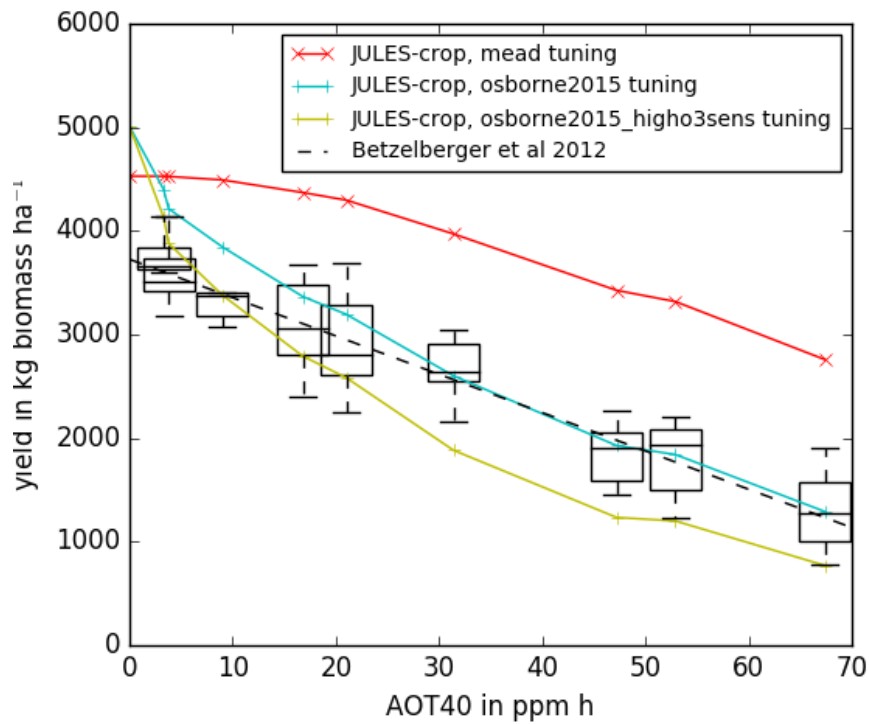

Figure 4 Black dashed line is the line of best fit from SoyFACE observation and the blue and green lines with crosses are the modelled output for each ozone concentration using the Osborne et al 2015 tuning with Sitch et al 2007 low and high sensitivity, respectively. The red line and crosses are the tuned parameters with Mead FLUXNET observation and SoyFACE ozone damage according to Table 4 and Figure 8.




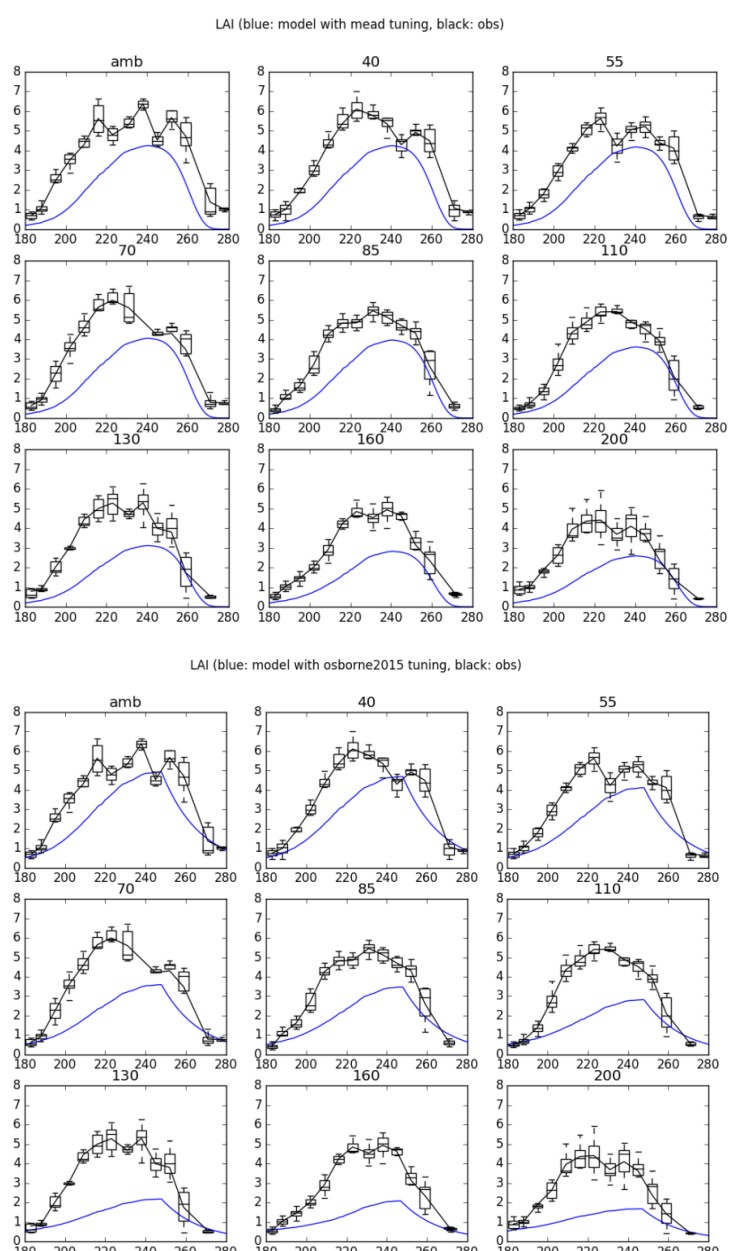

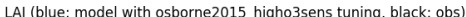

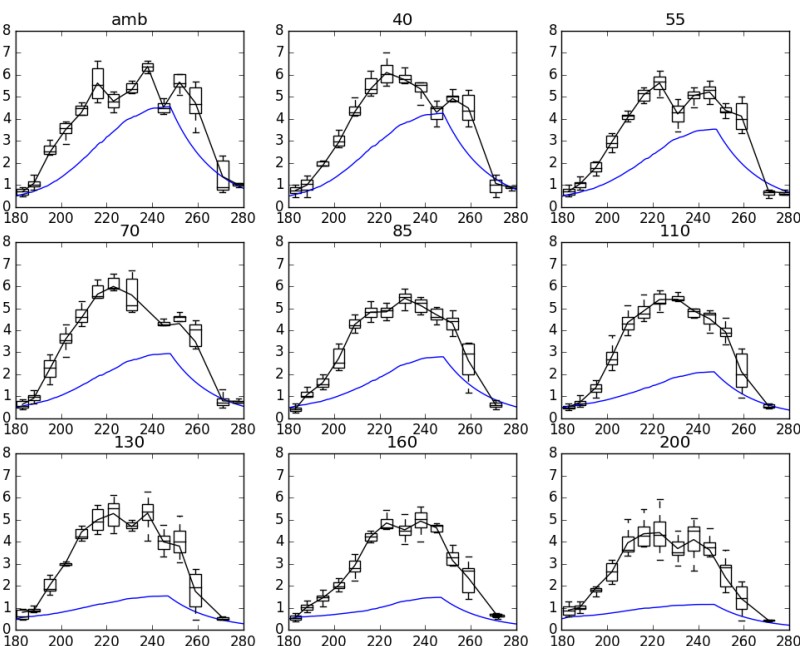

**Figure 5** Time series of Leaf Area Index (LAI) responses on different target ozone concentration at SoyFACE.
Black line is observed LAI from Betzelberger et al., (2012) and the blue line is JULES-crop LAI. Top:
calibrated JULES-crop using Mead observations. Bottom row: Osborne 2015 tuning with low sensitivity
(middle) and high sensitivity to ozone (bottom).








**Table 1.** JULES modules switches, which F (False) means turned off and T (True) means turned on. Asterisk indicates parameter was hard-wired and the description of the parameters at the bottom

|  | Osborne et al., (2015) | This study | Discussion |
|---|---|---|---|
| can_rad_mod | 5 (6 was not available) | 6 | Recommended option for layered canopy in version 4.6 |
| l_irrig_dmd | F | T | Irrigation on |
| irr_crop | - | 0 | |
| l_trait_phys | F | F | |
| l_scale_resp_pm | F | T | |
| l_leaf_n_resp_fix | F | - | Bug fix, affects can_rad_mod=5 but not can_rad_mod=6 |
| l_prescsow | T | T | Sowing dates available |

| Parameters | Description |
|---|---|
| Canopy radiation model | Number 6 is Multi-layer approach for radiation interception following the 2-stream approach of Sellers et al. (1992). This approach takes into account leaf angle distribution, zenith angle, and differentiates absorption of direct and diffuse radiation. it has a decline of leaf N with canopy height. Additionally includes inhibition of leaf respiration in the light. including: Sunfleck penetration though the canopy. Division of sunlit and shaded leaves within each canopy level. A modified version of inhibition of leaf respiration in the light. exponential decline of leaf N with canopy height proportional to LAI, following Beer's law. |
| L_irrid_dmd | Switch controlling the implementation of irrigation demand code. |
| Irr_crop | Irrigation season (i.e. season in which crops might be growing on the gridbox) lasts the entire year. |
| l_trait_phys | Switch for using trait-based physiology. Vcmax is calculated based on parameters nl0 (kgN kgC-1) and neff. |
| l_scale_resp_pm | Soil moisture stress reduces leaf, root, and stem maintenance respiration. |
| l_leaf_n_resp_fix | Switch for bug fix for leaf nitrogen content used in the calculation of plant maintenance respiration. |
| l_prescsow | Sowing dates prescribed |






**Table 2.** Parameter values in JULES-crop that are used to represent soybean. Asterisk indicates parameter
545         was hard-wired.

| | | Osborne et al., (2015) | This study | Discussion |
|---|---|---|---|---|
| $T_b$ | Base temperature (K) | 278.15 | 278.15 | Kept at Osborne et al., (2015) value |
| $T_o$ | Optimum temperature(K) | 313.15 | 313.15 | Kept at Osborne et al., (2015) value |
| $T_m$ | Maximum temp (K) | 300.15 | 300.15 | Kept at Osborne et al., (2015) value |
| $P_{sen}$ | Sensitivity of development rate to photoperiod (hours-1) | 0.0 | 0.0 | Kept at Osborne et al., (2015) value |
| $P_{crit}$ | Critical photoperiod (hours) | - | - | Not used when $P_{sen} = 0$ |
| $r_{dir}$ | coefficient determining relative growth of roots vertically and horizontally | 0.0 | 0.0 | Kept at Osborne et al., (2015) value |
| $\alpha_{root}$ | coefficient of partitioning to root | 20.0 | 19.8 | Supplementary Material 1.4.1 |
| $\alpha_{stem}$ | coefficient of partitioning to stem | 18.5 | 18.5 | Supplementary Material 1.4.1 |
| $\alpha_{leaf}$ | coefficient of partitioning to leaf | 19.5 | 19.2 | Supplementary Material 1.4.1 |
| $\beta_{root}$ | coefficient of partitioning to root | -16.5 | -15.47 | Supplementary Material 1.4.1 |
| $\beta_{stem}$ | coefficient of partitioning to stem | -14.5 | -13.195 | Supplementary Material 1.4.1 |
| $\beta_{leaf}$ | coefficient of partitioning to leaf | -15.0 | -14.287 | Supplementary Material 1.4.1 |
| $\gamma$ | coefficient of specific leaf area ($m_2$ kg-1) | 25.9 | 24.0 | Supplementary Material 1.4.3 |
| $\delta$ | coefficient of specific leaf area ($m_2$ kg-1) | -0.1451 | 0.15 | Supplementary Material 1.4.3 |
| $\tau$ | Remobilisation factor, fraction of stem growth partitioned to RESERVEC | 0.18 | 0.26 | Supplementary Material 1.4.3 |
| $f_{C,root}$ | Carbon fraction for dry root | 0.5 | 0.47 | Supplementary Material 1.4.4 |
| $f_{C,stem}$ | Carbon fraction for dry stem | 0.5 | 0.49 | Supplementary Material 1.4.4 |
| $f_{C,leaf}$ | Carbon fraction for dry leaf | 0.5 | 0.46 | Supplementary Material 1.4.4 |





| | | | | |
|---|---|---|---|---|
| $f_{C,harv}$ | Carbon fraction for harvest | 0.5 | 0.53 | Supplementary Material 1.4.4 |
| $\kappa$ | Allometric coefficient relating STEMC to CANHT | 1.6 | 1.9 | Supplementary Material 1.4.2 |
| $\lambda$ | Allometric coefficient relating STEMC to CANHT | 0.4 | 0.47 | Supplementary Material 1.4.2 |
| $\mu$ | Allometric coefficient for calculation of senescence | 0.05* | 5.0 | Supplementary Material 1.4.2 |
| $\nu$ | Allometric coefficient for calculation of senescence | 0.0* | 6.0 | Supplementary Material 1.4.2 |
| $DVI_{sen}$ | DVI at which leaf senescence begins | 1.5* | 1.25 | Supplementary Material 1.5 |
| $C_{init}$ | Carbon in crop at emergence in kgC/m2. | 0.01* | 3.5E-3 (Mead), 7.0E-3(SoyFACE) | Supplementary Material 1.4.5 |
| $DVI_{init}$ | DVI at which the crop carbon is set to initial carbon | 0.0* | 0.2 | Supplementary Material 1.4.5 |
| $T_{mort}$ | Soil temperature (second level) at which to kill crop if DVI>1 | t_bse_io* | 263.15 | Section 2.3 |
| $f_{yield}$ | Fraction of the harvest carbon pool converted to yield carbon | 1.0* | 0.74 | Section 2.3 |








**Table 3.** JULES plant functional type parameters extended to represent soybean. Asterisk indicates parameter
was hard-wired.

| | | Osborne et al., (2015) | This study | Discussion |
|---|---|---|---|---|
| $c3$ | c3_io | 1 | 1 | Soybean is a C3 plant. |
| $dr$ | rootd_ft_io | 0.5 | 0.5 | Not important in irrigated runs, so could not be tuned using US-Ne2 data. Kept at Osborne et al, (2015) value |
| $dq\mathrm{crit}$ | dq_crit_io | 0.1 | 0.1 | Kept at Osborne et al., (2015) value |
| $f_d$ | fd_io | 0.015 | 0.008 | Supplementary Material 1.4.6 |
| $f0$ | f0_io | 0.9 | 0.9 | Kept at Osborne et al., (2015) value |
| $n\mathrm{eff}$ | neff_io | $8.0\times 10^{-4}$ | $12.0\times 10^{-4}$ | Table 1 |
| $nl(0)$ | nl0_io | 0.073 | 0.1 | Table 1 |
| $T\mathrm{low}$ | tlow_io | 0.0 | 0.0 | Kept at Osborne et al., (2015) value |
| $T\mathrm{upp}$ | tupp_io | 36.0 | 36.0 | Kept at Osborne et al., (2015) value |
| $k_n$ | kn_io | 0.78 | - | Default for C3 grass for can_rad_mod5. |
| $k_{nl}$ | knl_io | - | 0.2 | Default for C3 grass for can_rad_mod6. |
| $Q10,\mathrm{leaf}$ | q10_leaf_io | 2.0 | 2.0 | Kept at Osborne et al., (2015) value |
| $\mu_{rl}$ | nr_nl_io | 1.0 | 0.390 | Supplementary Material S1-3 |
| $\mu_{sl}$ | ns_nl_io | 1.0 | 0.51 | Supplementary Material S1-3 |
| $rg$ | r_grow_io | 0.25 | 0.32 | Supplementary Material 1.4.6 |
| | orient_io | 0 | 0 | Kept at Osborne et al., (2015) value |
| $\alpha$ | alpha_io | 0.12 | 0.12 | Kept at Osborne et al., (2015) value |
| $\omega\mathrm{PAR}$ | omega_io | 0.15 | 0.15 | Kept at Osborne et al., (2015) value |
| $\alpha\mathrm{PAR}$ | alpar_io | 0.1 | 0.1 | Kept at Osborne et al., (2015) value |
| | fsmc_mod_io | 0 | 0 | Not important in irrigated runs, so could not be tuned using US-Ne2 data. Kept at Osborne et al (2015) value. |
| | fsmc_p0_io | 0.0 | 0.5 | FAO document 56 (Allen and Pereira, 2006) |
| $a$ | can_struct_a_io | 1.0 | 1.0 | Kept at Osborne et al., (2015) value |




**Table 4.** Summary of ozone parameter configurations employed in JULES-crop for the default Osborne et al., (2015) value and the tuned as calibrated to SoyFACE leaf gas-exchange measurements (note that these have been calibrated to daytime 8-hour concentrations and therefore will be different to parameters calibrated to monthly 24hour means)

| JULES ozone damage Parameters | Fractional reduction of photosynthesis by $O_3$ (sensitivity) (mmol m-2) (dfp_dcuo_io) | Threshold of ozone flux (mmol m-2 s-1) (fl_o3_ct_io) |
|---|---|---|
| Tuned value | 0.5 | 15.0 |
| Osborne et al 2015 (High sensitivity) | 5.0 | 1.4 |
| Osborne et al 2015 (Low sensitivity) | 5.0 | 0.25 |



**Appendix A**

Table A1 Summary of Land Surface Models (LSM) that contains crop tiles

| Crop model | Land Surface Model | Crops Function Type | Scale | Ozone damage on vegetation |
|---|---|---|---|---|
| LPJ-ml (managed land) | LPJ-GUESS (Bondeau et al., 2007) | 13 types of 11 arable crops and 2 managed grass types | Global | No |
| CLM-CROP | CESM (Community Land Model for Community Earth System Model) (Levis et al., 2011) | Four major crops: maize, soybean, rice and wheat | Global | Yes (Zhang et al., 2018) |
| Agro-IBIS | IBIS (Kucharik et al., 2003) | 12 natural and five crops: Maize, soybean, wheat, *Miscanthus giganteus*, sugarcane | Continental US | In development |
| ORCHIDEE-CROP | ORCHIDEE (Gervois et al., 2004; Berg et al., 2010, Wu et al., 2016) | Ten natural and Wheat, maize, soybean, sugarcane and more varieties | Europe and other regions | In development |
| JULES-Crop | JULES (Clark et al., 2011; Osborne et al., 2015) | 4 Crop Functional types, divided into C3 and C4. Potentially 12 crop functional types | Global | Yes. Flux-gradient approach to quantify $O_3$ interactive effects. |
| SIB | Simple Biosphere model (Lokupitiya et al., 2009) | Maize, soybean, wheat | North America | No |
| ISAM | Integrated Science Assessment Model (Song et al., 2013) | Maize-soybean rotation | North America | No |


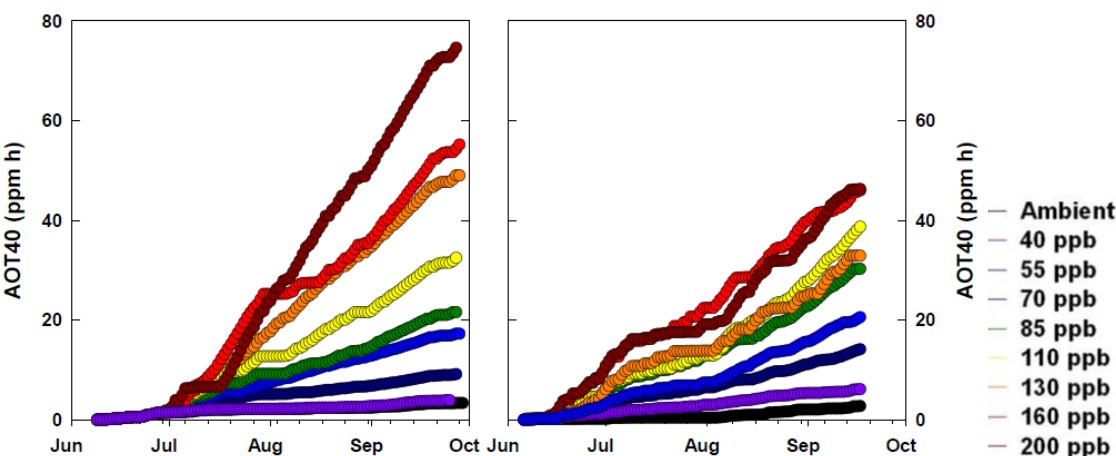

**Figure A2** AOT40 measured at SoyFACE in 2009 (left) and 2010 (right). With different target concentration.
(Betzelberger et al., 2012)