# Peer review of "Calibrating soybean parameters in JULES 5.0 from the US-Ne2/3 FLUXNET sites and the SoyFACE-O3 experiment"

_Geoscientific Model Development, 2020_

## Referee Comment (RC1) · Anonymous Referee #1 · 24 Jun 2020

-General comments The authors developed a crop-modelling framework considering the effect of ozone, which is one of the important climate change factors. The model performance was validated by the results obtained from a novel SoyFACE experiment. The manuscript addresses an important technical issue in the modelling how to incorporate with ozone impacts and what is a current problem for further development. The focus of the paper is well phrased and relevant for the journal Geoscientific Model Development. I have just a few minor comments and suggestions, that should be addressed before the final acceptance.

-Specific comments

[Figure]

L47: The authors mentioned "Recently. . ." but Morgan et al. 2004 is a quite old paper. Probably better to cite also latest ozone FACE studies (e.g. Agathokleous et al. 2017, Environmental Science and Pollution Research, vol. 24, pages 6634-6647; Paoletti et al., 2017, Science of the Total Environment, vol. 575, pages 1407-1414).

L147: Better to cite also CLRTAP (2017):

https://icpvegetation.ceh.ac.uk/sites/default/files/FinalnewChapter3v4Oct2017_000.pdf

L193-194, ". . . fractional reduction of photosynthesis by O3, F. . .", "F=1.40", "F=0.25": I suppose not "F" but "a" as you mentioned in lines 168-169.

L194, ". . .equation 1, 2 . . ..": I suppose not "equation 1, 2" but "equation 2, 3".

L221, "Threshold of O3 flux (mmol m-2 s-1)": The unit should be "nmol m-2 s-1".

L307: Better to add some brief sentence in order to support your speculation about plant density and leaf area. For example, Jaumer Ricaurte's paper (Ricaurte et al., 2016, Crop Science, vol. 56, pages 2713-2721) would be helpful.

Fig. 2: It is hard to identify symbols (model simulations vs observations). Better to use different colours as you did in the other figures.

Table 4: Correct the units. The unit of fractional reduction of photosynthesis should be "mmol-1 m2". Instead, the unit of the threshold of ozone flux should be "nmol m-2 s-1".

---

## Referee Comment (RC2) · Anonymous Referee #2 · 14 Sep 2020

Comments on:

**Calibrating soybean parameters in JULES 5.0 from the US- Ne2/3 FLUXNET sites and the SoyFACE-O₃ experiment**

Leung et al.,

submitted to Geoscientific Model Development, May 2020

Decision: revision and clarification needed before further consideration

General comments:

This manuscript calibrated physiological and other crop parameters for ozone damage in soybean in JULES-crop model, making use of measurements from the SoyFACE with ozone experiment and from FLUXNET sites. Then authors evaluated the model performance against yield data and LAI, and the model is able to reproduce reasonably good magnitude and seasonality (for LAI specifically). In general, I think this manuscript meets the criteria for publication on Geophysical Model Development:

- the manuscript contributes to ozone impact modeling

- scientific approach and methods used are valid, results are discussed an appropriate way, and resulted model could potentially be applied regionally and globally in future studies and could potentially help build a state-of-the-art impact assessment model

- modelling work is reproducible because of data and code availability, and sufficient description in the main manuscript and supplements

However, some questions and details need to be further addressed before further consideration, specifically:

- methods, results and conclusions should be presented in a clear way for the readers to follow

**Specific comments**:

- Could crop rotation and irrigation vs. rainfed issues affect the tuning parameters? Introduction and discussion on this are necessary for readers unfamiliar with the sites and the tuning process. In the tuning process, seems like 2002 and 2004 (and 2006 and 2008?) are picked for tuning, why are these years selected? Is it because of data availability or other reasons? Please clarify.

- Section 2.1 description is too short. Could it be extended by two or more sentences with more details in the main manuscript?

- Figure 3, 4 and Section 3 include the major results of this manuscript, which is the evaluation of aboveground carbon and yield against SoyFACE observations and previous model results, and the new, calibrated run underestimates ozone impact significantly at most of the ozone levels. Authors argue that this is due to underestimation of water stress in the model and some testing has been done. Could authors make some assumptions about water stress (like p0=0 mentioned) and include the results in Figure 3 and 4?

- Figure 5, could these figures be condensed into 9 panels or fewer instead of 27? So that three sets of model runs could be compared against each other. Results and discussion around Figure 5 could be easier to comprehend if they are compared side to side.

**Technical corrections**:
- Line 138, please include definition of daytime hours.
- Line 183, linear -> linearly
- Line 183, photosynthetic rate A, if A will not be used in the manuscript, don't include it.
- Line 189, '(dfp_dcuo_io)' is this used later? If not, don't include it.
- Line 173 and line 194, what is this F? is it the same as f in equation 2, 3 and 6. I am confused. Line 193-195 doesn't make sense to me.
- Line 191, should be "… the threshold ozone flux **above** which ozone would cause damage to …"
- Line 246, Section 3.1 is not necessary if there is not Sect. 3.2, 3.3, … and next section should be Section 4, instead of 5. Numbering in Sect. 2 has some issues too, please correct them.

**Issue with figures:**
- Simulation names: "Mead tuning", "Osborne 2015 tuning" and "Oseborne 2015 higho3sens tuning" in the figures and in the main text. Could these names be shortened and renamed?

- Supplementary Figure S1-1, S1-2, S3-2, S3-3 and S3-5, image quality is low (S1-1, low resolution), and presentation is not quite clear. Vertical and diagonal crosses are difficult to differentiate. Caption for Figure S1-2, should be 'Figure S1-1' instead of 'Figure 10'.

- Figure 4 caption: "… according to Table 4 and Figure 8." There is no Figure 8.

- Figure 3 title unnecessary. Titles and axis labels in other figures are also messy, these need to be fixed for readers to follow.

- Should Figure A2 be included in the supplements instead of the main manuscript? I don't see the necessacity of having appendix and supplements at the same time.

---

## Author Comment (AC1) · 25 Sep 2020

Author Responses to Reviewer 1 Comments on "Calibrating soybean parameters in JULES5.0 from the US-Ne2/3 FLUXNET sites and the SoyFACE-O3 experiment" by Leung et al., (Manuscript ID: gmd-2020-97-RC1)

Our point-by-point responses are provided below. The referees' comments are *italicized*, texts from the manuscript is in blue and our new/modified text is highlighted in **bold**. The revised manuscript with tracked changes is also included in the linked file below for the Editor's easy reference:

Response to Reviewer #1

We thank the reviewer for the complement and helpful comments. The paper has been revised substantially to address the reviewer's concerns point by point, and all changes are cited and discussed in the responses below.

*L47: The authors mentioned "Recently. . ." but Morgan et al. 2004 is a quite old paper.*

> Yes, I agree that Morgan et al., 2004 is an old paper. Thank you for suggesting other more recent publications. I have now included them.

> Recently the introduction of Free-Air-Concentration-Enrichment (FACE) technology avoids the artefacts from enclosed chambers, and $O_3$ fumigation was adapted to FACE facilities **(Agathokleous et al., 2017; Paoletti et al., 2017).**

*L147: Better to cite also CLRTAP (2017):*
*https://icpvegetation.ceh.ac.uk/sites/default/files/FinalnewChapter3v4Oct2017_000.pdf*

> Thanks for the suggestions. I have now updated it here at L148:

> To improve these indices, the Stockholm Environment Institute developed the Deposition of Ozone for Stomatal Exchange model ($DO_3SE$) **(Emberson et al., 2007; ICP Vegetation, 2017).**

*L193-194, ". . . fractional reduction of photosynthesis by O3, F. . .", "F=1.40", "F=0.25": I suppose not "F" but "a" as you mentioned in lines 168-169.*
*L194, ". . .equation 1, 2 . . ..": I suppose not "equation 1, 2" but "equation 2, 3".*

> Yes, you are correct. Sorry for the mistakes. Now I have change it in Line 195:

> …plant functional types with two different $O_3$ sensitivities (fractional reduction of photosynthesis by $O_3$**, *F*, *equation 2, 3*), where *a* = 1.40 is high sensitivity, and *a* = 0.25 is lower sensitivity** for C3 grass (Sitch, 2007), using monthly average $O_3$ data and calibration to yield observations.

*L221, "Threshold of O3 flux (mmol m-2 s-1)": The unit should be "nmol m-2 s-1".*

Yes, you are correct. I have amended it in L222:

We then tuned the $O_3$ parameterisation of Fractional reduction of photosynthesis by $O_3$ (sensitivity) and Threshold of $O_3$ flux (**nmol m-2 s-1**) to match the modelled leaf photosynthesis rate to the observed rate (Figure 2). The tuned parameters are showed in Table 4.

*L307: Better to add some brief sentence in order to support your speculation about plant density and leaf area. For example, Jaumer Ricaurte's paper (Ricaurte et al., 2016, Crop Science, vol. 56, pages 2713-2721) would be helpful.*

Thanks for the suggestions! This paper is very helpful. I have added a sentence in L310 to support the leaf area results.

Ricaurte et al., (2016) showed that higher sowing density would increase phyllochron in a linear relationship, which results a higher LAI measured that is consistent with our study.

*Fig. 2: It is hard to identify symbols (model simulations vs observations). Better to use different colours as you did in the other figures.*

Yes, I agree. I have now assigned colours for each symbol in Figure 2.

[Figure]

Figure 2. Net leaf $CO_2$ assimilation rate for calibrated JULES, simulated using the Leaf Simulator (black crosses) and observations from Betzelberger et al., (2012) (grey circles). X-axis is the daytime 8-hour mean $O_3$ concentration (ppb)

*Table 4: Correct the units. The unit of fractional reduction of photosynthesis should be "mmol-1 m2". Instead, the unit of the threshold of ozone flux should be "nmol m-2 s-1".*

Thanks for spotting the error. I have now corrected Table 4.

---

## Author Comment (AC2) · 25 Sep 2020

Author Responses to Reviewer 2 Comments on "Calibrating soybean parameters in JULES5.0 from the US-Ne2/3 FLUXNET sites and the SoyFACE-O3 experiment" by Leung et al., (Manuscript ID: gmd-2020-97-RC1)

Our point-by-point responses are provided below. The referees' comments are *italicized*, the texts from the manuscript are in blue and our new/modified text is highlighted in **bold**. The revised manuscript with tracked changes is also included in the linked file below for the Editor's easy reference:

Response to Reviewer #2

We thank the reviewer for the complement and helpful comments. The paper has been revised accordingly to address the reviewer's concerns point by point, and all changes are cited and discussed in the responses below.

*Could crop rotation and irrigation vs. rainfed issues affect the tuning parameters? Introduction and discussion on this are necessary for readers unfamiliar with the sites and the tuning process. In the tuning process, seems like 2002 and 2004 (and 2006 and 2008?) are picked for tuning, why are these years selected? Is it because of data availability or other reasons? Please clarify.*

> No, crop rotation and irrigation would not affect the tuning. We only simulated the years where soybean is grown. Maize grown in the odd years (2003, 2005 etc.) are not included in our tuning. JULES could simulate irrigation and refed and represent them well. Examples of JULES-crop representing irrigated and rainfed tunings can be found in Williams et al., (2017) Evaluation of JULES-crop performance against site observations of irrigated maize from Mead, Nebraska.

> The years 2004, 2006 and 2008 are picked because these are the years which soybeans are grown. I have clarified it in L104

> We first tuned the JULES-crop soybean parameterisation at the US-Ne2 and US-Ne3 Mead sites, where three years of soybean physiological and meteorological observations were available, at ambient ozone (Figure 1, steps 1-5). **The three years are 2004, 2006 and 2008 which soybeans were grown in Mead, maize were grown in other years.**

*Section 2.1 description is too short. Could it be extended by two or more sentences with more details in the main manuscript?*

> Yes, I agree that it is a bit too short. I have now extended it on L123

> Step 1 involved using Mead observation to tune the parameters needed by all PFTs in JULES with the crop model switched off. Step 2 is to evaluate the model

performance of GPP using Mead meteorology and LAI. Step 3 tunes the parameters needed by crop only. Step 4 evaluated the JULES-crop run performance with observed carbon pools in leaf, stem, harvest etc. Step 5 demonstrated the full JULES-crop runs at Mead using Mead meteorology and compared the model with observed GPP, aboveground carbon etc. Step 6 tune ozone damage using SoyFACE LiCOR measurements. And finally step 7 evaluates JULES-crop performance using SoyFACE meteorology and compare with observed yield and LAI.

*Figure 3, 4 and Section 3 include the major results of this manuscript, which is the evaluation of aboveground carbon and yield against SoyFACE observations and previous model results, and the new, calibrated run underestimates ozone impact significantly at most of the ozone levels. Authors argue that this is due to underestimation of water stress in the model and some testing has been done. Could authors make some assumptions about water stress (like p0=0 mentioned) and include the results in Figure 3 and 4?*

Yes, for the p0 value, we used the FAO document 56 (Allen and Pereira, 2006) which used the value p0=0.5 and I showed it in Table 3. I have now showed the results of p0=0 in Supplementary instead to avoid confusion. L262 is now updated

We tested the sensitivity to this choice by re-running this configuration with fsmc_p0=0 which represents water stressed conditions, and this caused a 12% reduction in aboveground carbon (plots show in Supplementary).

[Figure]

Supplementary Figure that shows p0=0 reduce the approximately 12% of the aboveground carbon and yield compared to Figure 3 and 4.

*Figure 5, could these figures be condensed into 9 panels or fewer instead of 27? So that three sets of model runs could be compared against each other. Results and discussion around Figure 5 could be easier to comprehend if they are compared side to side.*

Yes, you are right. I have condensed them into 9 panels, with each model set in different colours for easier comparison.

[Figure]

**Figure 5** Time series of Leaf Area Index (LAI) responses on different target ozone concentration at SoyFACE. Black line is observed LAI from Betzelberger et al., (2012) and the other lines are JULES-crop LAI with different tunings. Blue: calibrated JULES-crop using Mead observations. Green: Osborne 2015 tuning with low sensitivity. Red: Osborne 2015 tuning with high sensitivity to ozone.

*Line 138, please include definition of daytime hours.*

It is 12 hours between 0700 to 1900. It is now changed in L146

**The integral is taken over daytime hours between 0700 to 1900**

*Line 183, linear -> linearly*
*Line 183, photosynthetic rate A, if A will not be used in the manuscript, don't include it.*

Thanks, it is now changed in L191

…Given that $g_l$ and photosynthetic rate are **linearly** related [Cox et al., 1999], $g_l$ is given by…

*Line 189, '(dfp_dcuo_io)' is this used later? If not, don't include it.*

It is now changed in L196

Fractional reduction of photosynthesis with the instanteneous uptake of $O_3$ by leaves (mmol m$^{-2}$) determines the sensitivity of soybean to $O_3$

*Line 173 and line 194, what is this F? is it the same as f in equation 2, 3 and 6. I am confused. Line 193-195 doesn't make sense to me.*

Sorry for the confusions. I have change the letter $F$ to be capital letter to make it consistent. Please check L194

$$g_l = g_p F \qquad (6)$$

Where $g_p$ is the leaf conductance in the absence of $O_3$ effects. The set of equations (3,5,6) produces a quadratic relationship as a function of $F$, that can be solved analytically (Sitch et al., 2007).

*Line 191, should be "… the threshold ozone flux above which ozone would cause damage to …"*

Thanks for spotting the mistake. It is now changed in L196

Fractional reduction of photosynthesis with the instanteneous uptake of $O_3$ by leaves (mmol m$^{-2}$) determines the sensitivity of soybean to $O_3$ and the PFT-specific $O_3$ critical level (FO$_3$ crit) determines the threshold $O_3$ flux **above** which would cause damage to photosynthesis (Oliver et al., 2018; Sitch et al., 2007).

*Line 246, Section 3.1 is not necessary if there is not Sect. 3.2, 3.3, … and next section should be Section 4, instead of 5. Numbering in Sect. 2 has some issues too, please correct them.*

Thanks. I have corrected the numbering in section 2. And I have deleted the title of section 3.1. Please see L254

Results from JULES runs with crop model and ozone damage turned on are showed in Figure 3 and 4.

*Simulation names: "Mead tuning", "Osborne 2015 tuning" and "Oseborne 2015 higho3sens tuning" in the figures and in the main text. Could these names be shortened and renamed?*

Yes, it is now shortened to "mead", "Osborne 2015", "highO$_3$sen"

*Supplementary Figure S1-1, S1-2, S3-2, S3-3 and S3-5, image quality is low (S1-1, low resolution), and presentation is not quite clear. Vertical and diagonal crosses are difficult to differentiate. Caption for Figure S1-2, should be 'Figure S1-1' instead of 'Figure 10'.*

Thanks for the comments. We have plotted a higher resolution images in PDF format, I will attached it separately. And the caption for Figure S1-2 is changed in L44

Colours show the cosine for the zenith angle (for legend, **see Figure S1-1).** Solid black line indicates $a = 1$.

*Figure 4 caption: "... according to Table 4 and Figure 8." There is no Figure 8.*

Thanks. Figure 4 caption is now changed in L524

The red line and crosses are the tuned parameters with Mead FLUXNET observation and SoyFACE ozone damage according to **Table 4**.

*Figure 3 title unnecessary. Titles and axis labels in other figures are also messy, these need to be fixed for readers to follow.*

Sorry about that, I have fixed Figure 3.

*Should Figure A2 be included in the supplements instead of the main manuscript? I don't see the necessacity of having appendix and supplements at the same time.*

Yes, I agree. I have put the appendix in the supplementary materials.